# Sequence-to-Segments Networks
# for Segment Detection

**Zijun Wei[1]    Boyu Wang[1]    Minh Hoai[1]    Jianming Zhang[2]    Zhe Lin[2]**
**Xiaohui Shen[3]    Radomír Měch[2]    Dimitris Samaras[1]**
[1]Stony Brook University,    [2]Adobe Research,    [3]ByteDance AI Lab

## Abstract

Detecting segments of interest from an input sequence is a challenging problem which often requires not only good knowledge of individual target segments, but also contextual understanding of the entire input sequence and the relationships between the target segments. To address this problem, we propose the Sequence-to-Segments Network ($S^2N$), a novel end-to-end sequential encoder-decoder architecture. $S^2N$ first encodes the input into a sequence of hidden states that progressively capture both local and holistic information. It then employs a novel decoding architecture, called Segment Detection Unit (SDU), that integrates the decoder state and encoder hidden states to detect segments sequentially. During training, we formulate the assignment of predicted segments to ground truth as the bipartite matching problem and use the Earth Mover's Distance to calculate the localization errors. Experiments on temporal action proposal and video summarization show that $S^2N$ achieves state-of-the-art performance on both tasks.

## 1   Introduction

We address the problem of detecting temporal segments of "interest" in an input time series. Here we define "interest" as an abstract concept that denotes the parts of the data that have the highest (application dependent) semantic values. We assume there are training time series with annotated segments of interest (e.g., labeled by humans), and our goal is to train a neural network that can detect the segments of interest in unseen time series. This general problem arises in many situations including temporal event detection [17, 18], video summarization [47, 48], sentence chunking [32], gene localization [24], and discriminative localization [19, 31]. For human action detection, the segments of interest are the ones that correspond to the temporal extents of human actions. For video summarization, the segments of interest are the video snippets that summarize the video.

A typical approach to address this problem is to train a classifier to separate the annotated segments of interest from some negative examples. Once trained, the classifier can be used to evaluate individual candidate segments of the input time series in a sliding window approach to identify the segments of interest. This approach however has two drawbacks. First, the computational complexity depends on the number of candidate segments, and this scales quadratically with the length of the time series. Second, the independent evaluation of each segment is suboptimal for many situations because "interest" might be a contextual concept. To detect a set of target segments, not only do we need to evaluate the local content of individual segments, but also their collective relationships and their roles in the global context. Taking video summarization as an example, to summarize a video, it is important to know and preserve the gist of the video, and this requires a holistic analysis of the video. Furthermore, the set of selected video snippets should not overlap temporally or semantically, and this can only be avoided by collectively evaluating the segments. The second drawback of the sliding window classification approach is commonly addressed by applying a post-processing step such as

non-maximum suppression, but the addition of post processing steps creates a pipeline that cannot be optimized end-to-end.

In this paper we propose the Sequence-to-Segments Network ($S^2N$), a novel recurrent neural network for analyzing a time series to detect temporal segments of interest. Our network is based on the sequential encoder-decoder architecture [40]. The encoder network encodes the time series and produces a sequence of hidden states that progressively capture from local to holistic information about the times series. The decoder network takes the final state of the encoder network as its starting state and outputs one segment of interest at a time. The state will be updated to incorporate what has been already outputted. This alleviates the need for a post-processing step that may not have access to the time series information. The whole encoder-decoder pipeline can be optimized end-to-end. For the decoder network, we introduce a novel architecture, named Segment Detection Unit (SDU), which outputs a segment based on the decoding state and the hidden states of the encoder. The SDU localizes the segment of interest by pointing to the boundaries of the segment, similar to the pointer network [43]. The SDU also outputs a confidence value for the selected segment. The computational complexity of SDU is linear with respect to the length of the input sequence, which is more efficient than the quadratic complexity of the sliding window approach.

To train an $S^2N$, we optimize a loss function that is defined based on the localization offsets and the recall rate of the proposed segments. This loss function is computed based on the minimum matching cost between the target segments of interest and the sequence of detected segments [13, 39]. Inspired by [39], we use a lexicographic comparison function for the detection-target pairs and use the Hungarian algorithm to find the best matching. In addition, we use the Earth Mover's Distance loss that accounts for the localization error to train the boundary pointing modules of the SDU.

The major contributions of this paper are: (1) We propose $S^2N$, a novel network architecture for detecting segments of interest in video. (2) We design a matching algorithm and an Earth Mover's Distance based loss function for the training of $S^2N$s. (3) We show that $S^2N$s outperform the state-of-the-art methods in two real-world applications: human action proposal and video summarization.

## 2 Related Work

Recurrent Neural Networks (RNNs) have been the standard method for learning functions over sequences from examples for a long time [34]. To further remove the constraint that the number of outputs is dependent on the number of inputs, Sutskever *et al.* [40] recently proposed the sequence-to-sequence paradigm that first uses one RNN to map an input sequence to a state and then applies another RNN to output a sequence with arbitrary length based on the encoded state. Bahdanau *et al.* augmented the decoder by propagating extra contextual information from the input using a content-based attentional mechanism [1, 14]. Vinyals *et al.* [43] modified the attention model to allow the model to directly point to elements in the input sequence, providing a more efficient and accurate model for element localization. These developments have made it possible to apply RNNs to new domains such as language translation [1, 40] and parsing [44], and image and video captioning [7, 45]. However, the current RNNs are designed to output each time one "token" in the input sequence, they can not handle properly the segment detection task in which each time a continuous chunk of the inputs is selected. Perhaps the most related work to ours is [13] which attempts to train RNNs to label unsegmented sequences directly. But the goal of [13] is classification where the localization information is not required in the output. The proposed $S^2N$ simultaneously detects segments and estimate their confidence scores, thus can be applied to different problems such as temporal action proposal generation and video summarization.

## 3 Sequence-to-Segments Network Architecture

In this section, we will describe the $S^2N$. We first formally state the problem. We then describe the overall $S^2N$ architecture and the details of the proposed Segment Detection Unit (SDU), the core component of $S^2N$ for localizing a temporal segment of interest.

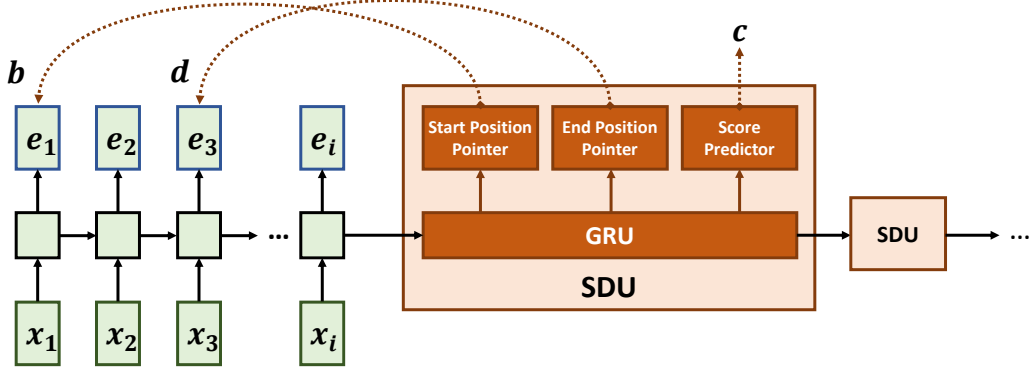

Figure 1: An Encoder (green) processes the input sequence to create a set of encoding vectors ($\{e_1, e_2, ...e_M\}$). At each decoding step, a Segment Detection Unit (SDU) updates the decoding state with a GRU, and based on the updated state, the SDU points to the beginning (**b**) and ending positions (**d**) with two separate pointing modules and estimates the confidence score (**c**) of the segment.

## 3.1 Problem Formulation

Let $X = (\mathbf{x}_1, \mathbf{x}_2, \cdots, \mathbf{x}_M)$ be an input time series of length $M$, where $\mathbf{x}_m \in \mathcal{R}^d$ is the observation feature vector at time $m$. Our goal is to learn an RNN that can localize a set of segments of interest $\mathcal{S} = (S_1, \cdots, S_N)$ from the input time series $X$. Here each segment $S_n$ corresponds to a contiguous subsequence of $X$ and it is parameterized by a tuple of three elements $(b_n, d_n, c_n)$ indicating the beginning position $b_n$, the ending position $d_n$, and the estimated interest score $c_n$. There are no explicit constraints on the locations and extents of the segments; the segments can overlap and their union does not have to cover the entire sequence $X$. Intuitively, many problems that detect temporal segments in a series such as action detection or video summarization can be formulated this way.

## 3.2 Model Overview

The proposed S$^2$N is illustrated in Fig. 1. S$^2$N is a sequential encoder-decoder with an attentional mechanism [1]. S$^2$N sequentially encodes an input sequence $\mathbf{x}_1, \cdots, \mathbf{x}_M$ and obtains a corresponding sequence of encoding state vectors $\mathbf{e}_1, \cdots, \mathbf{e}_M$; the encoding state vector $\mathbf{e}_m$ essentially contains integrated information from $\mathbf{x}_1$ to $\mathbf{x}_m$ [23, 40].

## 3.3 Segment Detection Unit (SDU)

A key component of the S$^2$N is the Segment Detection Unit (SDU) for localizing a segment of interest. As shown in Fig 1, each SDU has four components: a Gated Recurrent Unit (GRU) [5] for updating and communicating states between time steps, two pointing modules [43] for pointing to the beginning and ending positions of the segment, and a score estimator for evaluating the interest score of the segment. Details about these components are described below.

**GRU for state update.** During decoding, at each step given the previous hidden state $\mathbf{h}_{j-1}$ ($\mathbf{h}_0$ is the concatenation of the last hidden state and memory cell of the encoder), the GRU module updates the current hidden state: $\mathbf{h_j} = \text{GRU}(\mathbf{h}_{j-1}, \mathbf{z})$, where $\mathbf{z}$ is a learned input vector to the GRU at each step. We refer the reader to [5] for further details about the GRU update function.

Note that S$^2$N can be theoretically used with any RNN architecture, including LSTM, GRU, and their variants (e.g., [26]). We propose to use GRU [5] because it has a simpler architecture and fewer parameters than the others (which means higher training and testing efficiency). We also experimented with LSTM but did not observe significant difference in terms of model accuracy. This is consistent with prior observations [3] and empirical findings from prior work on deep recurrent models in other domains [5, 6, 22].

**Pointing modules for boundary localization.** Given the current state $\mathbf{h}_j$ of an SDU, we predict the two boundary positions similar to the pointer networks (Ptr-Net) [43]. To localize the beginning

position $b_j$ of a segment, we use the pointer mechanism as follows:

$$b_j = \underset{i}{\operatorname{argmax}}\, g(\mathbf{h}_j, \mathbf{e}_i), \text{where } g(\mathbf{h}_j, \mathbf{e}_i) = \mathbf{v^T}\tanh(\mathbf{W_1 e_i} + \mathbf{W_2 h_j}). \qquad (1)$$

The beginning boundary is determined as the location that has the highest response to a pointer function $g$. The output of this function depends on the state $\mathbf{h}_j$ of the SDU and the encoding vector $\mathbf{e}_i$ of the encoder component.

One alternative of predicting the locations is to use regression (similar to [27, 33]), however, this approach outputs a ratio in $[0, 1]$, which does not respect the constraint that the outputs map back exactly to the boundaries. As demonstrated in prior works [38, 43], the predictions are blurry over longer sequences.

Note the difference compared to the original Ptr-Net [43]: the pointer function is defined based on the encoding state vector $\mathbf{e}_i$ instead of the input vector $\mathbf{x}_i$. The encoding state vector $\mathbf{e}_i$ contains richer information than the input vector $\mathbf{x}_i$; $\mathbf{e}_i$ integrates the progression of the input time series up until time $i$, and this information is crucial for determining the segment boundaries [29]. In the above, $\mathbf{v}$, $\mathbf{W}_1$ and $\mathbf{W}_2$ are learnable parameters of the pointing module that associates the decoding state with the hidden encoding states.

Similarly, the ending position $d_j$ is determined using another independent Ptr-Net module. Thus, we have two Ptr-Net modules for determining the locations of the beginning and ending positions.

**Score predictor**. Finally, we estimate the confidence score of the segment using a two layer 1D convolution network with a ReLu activation layer in between.

**No terminal output.** We do not design a terminal output for S$^2$N as in [1] because of two reasons. First, the problem we address is to output a ranked list of temporal segments of interest, which is different from the problem of sequence-to-sequence translation, in which there is a need for a terminal state. Second, by not having a terminal state, S$^2$N can output as many segments as needed, bringing flexibility to different needs in real-world problems.

## 4 Training a Sequence-to-Segments Network

The S$^2$Ns can be trained end-to-end. In this section, we first present the loss function, and then describe how we match the sequence of predicted segments to the set of target segments.

Let $\mathcal{G} = \{G_1, \cdots, G_K\}$ denote the set of ground truth segments and $\mathcal{S} = (S_1, \cdots, S_N)$ the sequence of segments produced by the S$^2$N. Given an assignment strategy for matching $\mathcal{G}$ to $\mathcal{S}$, we will have an injection mapping: $f : \{1, \cdots, K\} \to \{1, \cdots, N\}$, where $f(k)$ indicates that the ground truth instance $G_k$ should be matched to $S_{f(k)}$. Then, the loss value for the predicted sequence of segments and the set of ground truth instances is computed as follows:

$$\mathcal{L}(\mathcal{G}, \mathcal{S}, f) = \alpha \sum_{k=1}^{K} \mathcal{L}_{loc}(G_k, S_{f(k)}) + \sum_{n=1}^{N} \mathcal{L}_{conf}(S_n, \delta_n), \qquad (2)$$

where $\delta_n$ is the desired confidence value for $S_n$ (depending on whether $S_n$ is matched to a ground truth instance in $\mathcal{G}$). $\mathcal{L}_{loc}$ and $\mathcal{L}_{conf}$ are the loss functions for localization and confidence score prediction, which will be explained below.

**Loss function for localization.** For a given probability distribution over the location of the segment boundary returned by the pointing module, one way to define the localization loss is to use the cross-entropy loss as in [43]. However, this loss function is unsuitable for boundary localization because it is insensitive to the amount of localization error; this loss function does not provide meaningful gradients for the training process.

We propose to use a loss function that is defined based on the Earth Mover's Distance (EMD) between the probability distribution of the predicted boundary and the distribution that represents the ground truth boundary. We now explain how this loss function can be computed for the beginning position $b$ (the loss for the ending position $d$ is computed similarly). Recall from Eq. (1) that we determine the beginning location of a segment as the maximum of a response function: $b = \operatorname{argmax}_i g(\mathbf{h}, \mathbf{e}_i)$, where $\mathbf{h}$ is the state vector of the SDU. We define the probability of picking $i$ as the boundary point

based on the soft-max function $Pr(b = i) = \exp(g(\mathbf{h}, \mathbf{e}_i)) / \sum_i \exp(g(\mathbf{h}, \mathbf{e}_i))$. Let $\mathbf{p}^*$ be the binary indicator vector for the ground truth location of segment boundary; $p^*(i) = 1$ if $i$ is the ground truth boundary and 0 otherwise. The EMD loss can be computed based on the differences the two cumulative distributions:

$$\mathcal{L}_{loc}^b = \sum_{m=1}^{M} \left( \sum_{i=1}^{m} Pr(b = i) - \sum_{i=1}^{m} p^*(i) \right)^2. \tag{3}$$

We use the squared loss in Eq. (3) because it usually converges faster than a $L_1$ loss and is easier to optimize with gradient descent [20, 28, 35]. The loss for the predicting the ending position is similarly defined and the total localization loss is: $\mathcal{L}_{loc} = \mathcal{L}_{loc}^b + \mathcal{L}_{loc}^d$.

**Loss function for confidence estimation.** Recall that the $S^2N$ predicts a confidence value $c_n$ for each segment $S_n$. We can use the cross-entropy loss to measure the compatibility between $c_n$ and $\delta_n$: $\mathcal{L}_{conf}(S_n, \delta_n) = -\delta_n \log(c_n) - (1 - \delta_n) \log(1 - c_n)$. For some applications, such as video summarization, the desired confidence value for each segment $S_n$ is not necessary binary. In this case, we can use the $L_2$ loss function, i.e., $\mathcal{L}_{conf}(S_n, \delta_n) = (c_n - \delta_n)^2$.

**Assignment Strategy.** To implement the above loss functions, we need an assignment strategy to match the target segments to the predicted ones. We follow the bipartite matching strategy based on the Hungarian loss used in [39]. Specifically, we define the matching cost between a predicted segment $S_n$ and a ground truth $G_k$ using a triplet cost function:

$$\Delta(G_k, S_n) = (o_{kn}, n, l_{kn}). \tag{4}$$

The function $\Delta : \mathcal{G} \times \mathcal{S} \to \Re^3$ returns a tuple where $l_{kn}$ is the $L_1$ distance between $G_k$ and $S_n$. $o_{kn}$ indicates whether there is significant overlapping between $G_k$ and $S_n$:

$$o_{kn} = \begin{cases} 1 & \text{if } IoU(G_k, S_n) \geq 0.5 \\ 0 & \text{otherwise.} \end{cases} \tag{5}$$

We can use the Hungarian algorithm to determine the best matching with lexicographic preference:

$$\sum_{k=1}^{K} \Delta(G_k, S_{f(k)}) = \left( \sum_{k=1}^{K} o_{kf(k)}, \sum_{k=1}^{K} f(k), \sum_{k=1}^{K} l_{kf(k)} \right). \tag{6}$$

In words, the Hungarian algorithm first finds the best matching based on $o$ only. For tie-breaking, it will consider $n$, and then $l$ if necessary. For more details, see [39].

## 5 Experiments

### 5.1 Model Implementation and Hyper-parameters

We used the same architecture in all experiments even though better results can likely be achieved by tuning the model to fit specific problems. Unless specified otherwise, the encoder is a 2 layer bi-directional GRU with 512 hidden units with dropout rate 0.5, the GRU module in SDU is one-directional with 1024 hidden units. All the models are trained with the Adam optimizer [25] for 50 epochs with an initial learning rate of 0.0001, which was decreased by a factor of 10 when the training performance plateaued, batch size of 32 and $L_2$ gradient clipping of 1.0. The trade-off factor $\alpha$ in Eq. (2) is set to ensure that $\mathcal{L}_{loc}$ does not dominate in the total loss. A weight adjustment for the score predictor is also used if necessary to account for the imbalance between the positive and negative samples. The code is publicly available at `https://www3.cs.stonybrook.edu/~cvl/projects/wei2018s2n/S2N_NIPS2018s.html`

### 5.2 Temporal Action Proposal

Temporal Action Proposal (TAP) generation, akin to generation of object proposals in images, is an important problem as accurate extraction of semantically important segments (e.g., human actions)

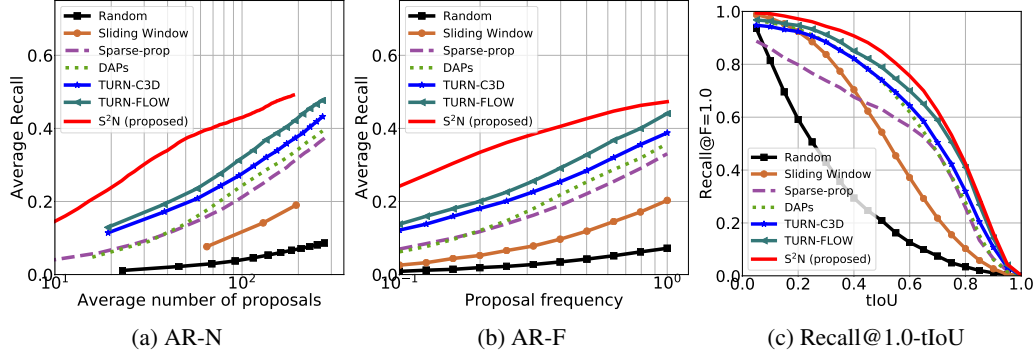

|             |             |                   |
|-------------|-------------|-------------------|
| (a) AR-N    | (b) AR-F    | (c) Recall@1.0-tIoU |

Figure 2: $S^2N$ outperforms previous temporal and temporal action proposal generation approaches on THUMOS-14 under various performance metrics.

from untrimmed videos is an important step for large-scale video analysis. In this section we show that an $S^2N$ can be trained to generate action proposals.

**Dataset.** We evaluate $S^2Ns$ on the THUMOS14 dataset [21], a challenging benchmark for the action proposal task. Following the standard practice, we train an $S^2N$ on the validation set and evaluate it on the testing set. On these two sets, 200 and 212 videos have temporal annotations in 20 classes, respectively. The average video duration in THUMOS14 is 233 seconds. The average number of labeled actions in each video is around 15. The average action duration is 4 seconds and more than 99% of the actions are within 10 seconds. We train an $S^2N$ using 180 out of 200 videos from the validation set and hold out 20 videos for validation.

**Implementation** For each video, we extract C3D features [41] following [3, 8]. To address the problem of long videos, we split each video into overlapping chunks of 360 frames (~12s) and subsample every 4 frames. We set the number of proposals generated from each chunk to be 15, which is the largest possible number of ground truth proposals contained in a chunk during training. We combine the proposals from chunks, sort them by their scores, and apply a Non-Maximum Suppression (NMS) with a 75% temporal intersection over union (tIoU). Note this is the **only** post-processing step used to address the overlap introduced in splitting the videos.

**Metrics**. We compare $S^2N$ with the baselines under the following metrics:

*AR-N* [46]: AR-N measures average recall (AR) as a function of number of proposals per video. Note that the numbers of retrieved proposals ($N$) for all the test videos are the same regardless of their lengths. Under this metric, we limit $N$ to 300 considering that on average each video only contains 15 actions.

*AR-F* [9]: AR-F measures average recall (AR) as a function of proposal frequency ($F$), which denotes the number of retrieved proposals per second for a video. For a video of length $L_i$ seconds and proposal frequency of $F$, the retrieved proposal number of this video is $N_i = F \times L_i$.

*Recall@F-tIoU* [9]: this metric measures the recall rate at proposal frequency $F$ with regard to different tIoUs. In the evaluation, we set $F = 1.0$ following [9].

**Baselines.** We compare $S^2N$ to the state-of-the art TAP generation methods including DAPs [8] that uses an encoder LSTM and a regression branch for localization, Sparse-prop [4] that applies dictionary learning for class independent proposal generation over a large set of candidate proposals, and TURN-TAP [9] that evaluates candidate proposals in a sliding window manner over different temporal scales and level of contexts (we compare with variants of TURN-TAP based on different features and denote them as TURN-C3D and TURN-FLOW). We also compare with *sliding window* and *random* generators. For the DAPs, Sparse-prop, and TURN-TAPS, we plot the curves using the generated proposals provided by the authors. The sliding window proposals and random proposals are generated following [9].

**Results.** The comparison to baselines under *AR-N*, *AR-F*, and *Recall@F=1.0-tIoU* metrics are shown in Fig 2. $S^2N$ outperforms the baselines by a significant margin over all the metrics. Note the gap between $S^2N$ and DAPs partially implies the necessity of considering the contextual information

Table 1: $F1$ scores (%) of various video summary methods on the SumMe dataset [15]

| Interesting[15] | Submodularity[16] | DPP-LSTM[47] | GAN$_{sup}$[30] | DR-DSN$_{sup}$ [48] | S$^2$N(proposed) |
|:---:|:---:|:---:|:---:|:---:|:---:|
| 39.4 | 39.7 | 38.6 | 41.7 | 42.1 | **43.3** |

and the superiority of the proposed pointing mechanism. Also note that we did not apply any post processing such as using the action length distributions as priors [9, 36], merging neighboring proposals or boundary refinement [9, 37] other than a simple non-maximum suppression step.

**Ablation Study.** We explore the influence of different label assignment strategies and loss functions on the performance of S$^2$N. Specifically we compare the proposed S$^2$N with the following variants:

*CLS-FIX*: optimize the localization errors using cross-entropy classification loss as suggested in [43] and assign labels to predictions base on a fixed order matching).

*CLS-HUG*: optimize the localization errors with cross-entropy loss and assign labels to predictions base on the Hungarian matching algorithm described in Sec. 4.

*EMD-FIX*: optimize the localization errors with the EMD loss as in Eq. (3) and assign labels based on the fixed order matching.

*L2-FIX /HUG*: optimize the localization errors with the $L_2$ loss as an alternative to EMD loss and assign labels based on the fixed order matching or the Hungarian matching algorithm.

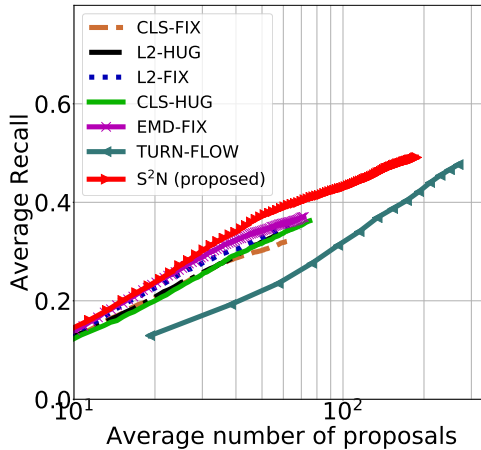

Figure 3: Comparing different action proposal methods. Best viewed on a digital device.

As shown in Fig. 3, the proposed strategy to train the S$^2$N significantly outperforms its variants. The variant methods tend to generate overlapping proposals so that the post-processing NMS reduces the effective number of proposals significantly.

**Speed.** S$^2$N is efficient since it does not require repeated computation over multi-scale context. Specifically, S$^2$N processes each frame in a sequence only once in the encoding stage and outputs a fixed set of segments over the whole sequence in the decoding stage. It is more efficient than recent models ( [2, 3]) that evaluate on a dense set of highly-overlapped candidates at each temporal step in a sequence. Quantitatively, it takes on average 0.028s to process a 12s, 30FPS video on a GTX Titan X Maxwell GPU with 12GB memory. In the batch mode, it takes around 2s to generate over 1200 proposals for an 8-minute video (14400 frames sampled every 4 frames). This is more than two times faster than the recently proposed models (1800 FPS *v.s.* 701 FPS [2] *v.s.* 308 FPS [3] *v.s.* 134 FPS [8]).

## 5.3 Video Summarization

Automatic video summarization provides a method for humans to browse and analyze video data. A good video summarization algorithm need to select a small set of segments that are interesting, diverse, and representative of the original video. In this section we show that S$^2$N can be trained to summarize long videos by generating a set of segments.

**Dataset.** We perform experiments on SumMe [15], a standard benchmark for video summarization. SumMe consists of 25 user videos covering various topics such as holidays and sports. Each video in SumMe ranges from 1 to 6 minutes and is annotated by 15 to 18 people (thus there are multiple ground truth summaries for each video). We treat each annotation separately and consider all of them ground truth. In this way, S$^2$N is trained to model multiple segment combinations to account for different user annotations (around 450 annotated video instances). We use the canonical setting suggested in [47] for evaluation: we use the standard 5-fold cross validation (5FCV), i.e., 80% of videos are for training and the rest for testing.

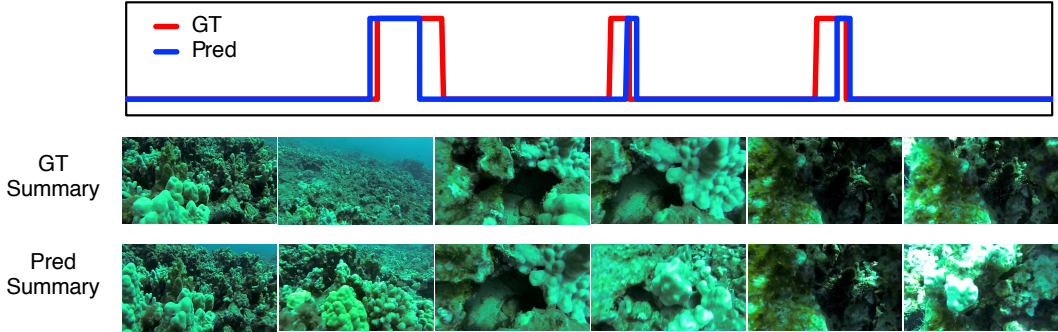

Figure 4: Visualization of the summarization results. $S^2N$ localizes the interesting events in the video preferred by the annotators.

**Implementation.** Similar to temporal action proposal generation, we use C3D features. Each video is split into overlapping chunks of 800 frames, subsampled every 8 frames as inputs. We limit the maximum number of output segments to 6. To generate a summary, following the standard practice [47, 48], we select segments based on their scores by maximizing the total scores while ensuring that the summary length does not exceed a limit, which is usually 15% of the video length. The maximization step is essentially the 0/1 Knapsack problem. To address the problem that SumMe has limited training data. We train each split for exactly 10 epochs and report the performance based on the last epoch.

**Evaluation metric.** We follow the commonly used protocol from [16, 47, 48]: we compute the F1-score to assess the similarity between the predicted segments and the ground truth summaries. To deal with the existence of multiple ground truth summaries [16], we evaluate the predictions w.r.t. the nearest-human summary, i.e., the one that is the most similar to the automatically created one.

**Baselines.** We compare $S^2N$ to multiple state-of-the-art video summary algorithms including interestingness-based summary [15], submodularity-based summary [16], and the recent deep learning based models, including: DPP-LSTM [47] (based on LSTM and a determinantal point processes [11]), $GAN_{sup}$ [30] ( based on GAN [12] with extra supervision), and $DR\text{-}DSN_{sup}$ [48] (based on reinforcement learning with supervision).

**Results.** As shown in Tab 1, $S^2N$ outperforms all other methods. $S^2N$ is designed to capture all the information needed for generating good summaries. We also visualize an example of the summarization in Fig 4.

# 6 Conclusions and Future Work

We have proposed the Sequence-to-Segments Network ($S^2N$), a novel architecture that uses Segment Detection Units (SDU) to detect segments sequentially from an input sequence. We have shown that $S^2N$ can be applied to real-world problems and achieve state-of-the-art performance.

There are a a few directions for future work. One direction is to augment the encoding stage to be capable of recording longer sequences [26]. Another possible direction is to extend $S^2N$ to more complex problems such as action detection in untrimmed videos. A third direction is to introduce auxiliary losses to enforce explicit semantic constraints on $S^2N$ [2]. It is also possible to base $S^2N$ on the fully convolutional encoder-decoder architecture [10, 42].

**Acknowledgements.** This project was partially supported by NSF-CNS-1718014, NSF-IIS-1763981, NSF-IIS-1566248, the Partner University Fund, the SUNY2020 Infrastructure Transportation Security Center, and a gift from Adobe.

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
