[Reviews · NeurIPS 2018]

Reviewer 1



The authors describe an architecture to output a series of temporal regions (segments) of interest in a video. There is a simple RNN encoder and an interesting decoder called a "Segment Detection Unit" which points to multiple segment boundaries. SDU is applied recurrently to the final state of the encoder, each iteration outputs a segment and a confidence score. (see End-to-End people detection, which is referenced [36]) SDU is similar to Pointer Networks in that it points back to an element of the input sequence as its output rather than outputing a regression value or an index to a dictionary. Also they 'improve' on pointer networks by using the encoded state of each input timestep (along with the current network state) (instead of the input at each timestep ) as input to the function that determines the pointer (function g). The encoder is thus able to learn to encode states with features to help learn g. So the model encodes the whole sequence, and the final state is input to a recurrent SDU unit, which has outputs for beginning and end. The SDU unit outputs for each output timestep are calculated for each encoded state of the input sequence, via a function g that uses the encoded states of the input sequence and the current decoder state as input) in this way the authors directly score each of the input frames as beginning and end segment positions and take the max of those scores to output a segment. They also learn to output a confidence score for each segment. They define a loss function that considers localization loss and confidence of predictions. For localization loss, they use the EarthMover distance. Their loss function depends on an assignment of all output segments to all ground truth segments and use the "Hungarian algorithm" to do just that. They perform reasonable experiments, compare with other methods, and show good performance. Concern: There is no terminal output state from the deocder RNN. In the Implementation section (201) it says: "We set the number of proposals generated from each chunk to be 15 which is the largest possible number of ground truth proposals contained in a chunk during training". Does this mean that 15 outputs are always made, but that the confidence score is used as a cutoff for accepting segments? It is not clear to me that this is explained. I am not saying this is bad, it's just not clear.. if my interpretation is true then what is the threshold used? How is it determined? This is a clear-headed work, combining ideas from recent advances. The choices made appear creative and solid, and the results are good. Minor points: This is not true (line 75): "However, the current RNNs are still restricted to output each time a “token” of the input sequence", but the following sentence is fine. Grammar 83: 'a 1D "object" or "objects"

Reviewer 2



The paper looks into the problem of detecing segments of interest in a time series. A novel network architecture for the detection is proposed, trained with a loss function based on the Earth Mover's Distance. The model is evaluated on videos for action proposal generation and video summarization. The goal of the research is very clear and the paper is well-written. It would be good to make the code available to be able to reproduce the results, since the actual implementation details and values are kept quite vague in the text. The proposed network architecture and loss function are interesting. The adventage of the new loss function over cross-entropy is demonstrated in the experiments. The choice of GRU, however, is not explicitly motivated. Testing the network on action proposal generation shows the flexibility of the method: it could be useful in multiple applications. The method is also tested for video summarization. In this case, only six segments are given as output per video. This sounds like manual tuning is needed to make sure the network outputs the desired results in a new application; simply retraining might not be sufficient. The results for both problems exceed state of the art. The strategy is only tested on video sequences (action proposals, summarization and supplemental material), while it is written in a very general way: 'time series'. It would therefore be good if some results could be presented for other applications, for instance speech recognition or EEG.

Reviewer 3



The paper presents a new method, S2N, to detect segments of interest in a video sequence. The authors propose an RNN-based architecture to predict the segment boundary. The authors, borrowing the idea from the pointer network, design the SDU, which predicts the pointers to the start, end positions and a confidence score of the segment based on the hidden states. The earth mover’s distance loss is proposed for training S2N. The proposed method achieves state-of-the-art performance on action proposal generation and video summarization. Pros: + The paper presents two novel components: 1) using the SDU and 2) applying the EMD loss on segment detection. + The method achieves state-of-the-art on two applications. Cons: - No careful ablation study: especially what is the performance of the method using the proposed loss as opposed to some other loss functions such as cross entropy or l2 loss. How does the assignment strategy influence the performance? - No the computational time comparison with the baselines. Minor: Repeated citation: [9] and [10] I lean to accept the paper with the novel contributions and the good results, but I would like to see the above concerns being addressed in the rebuttal. My main concerns are addressed in the author response. I recommend to accept the paper.